# Bacterial Contamination of Open Fractures: Pathogens and Antibiotic Resistance Patterns in East China

**DOI:** 10.3390/jpm13050735

**Published:** 2023-04-26

**Authors:** Wanrun Zhong, Yanmao Wang, Hongshu Wang, Pei Han, Yi Sun, Yimin Chai, Shengdi Lu, Chengfang Hu

**Affiliations:** Department of Orthopaedics, Shanghai Sixth People’s Hospital Affiliated to Shanghai Jiao Tong University School of Medicine, Shanghai 200235, Chinalushendi0828@163.com (S.L.)

**Keywords:** bacterial contamination, open fractures, pathogens, East China

## Abstract

Bacterial contamination of soft tissue in open fractures leads to high infection rates. Pathogens and their resistance against therapeutic agents change with time and vary in different regions. The purpose of this study was to characterize the bacterial spectrum present in open fractures and analyze the bacterial resistance to antibiotic agents based on five trauma centers in East China. A retrospective multicenter cohort study was conducted in six major trauma centers in East China from January 2015 to December 2017. Patients who sustained open fractures of the lower extremities were included. The data collected included the mechanism of injury, the Gustilo-Anderson classification, the isolated pathogens and their resistance against therapeutic agents, as well as the prophylactic antibiotics administered. In total, 1348 patients were included in our study, all of whom received antibiotic prophylaxis (cefotiam or cefuroxime) during the first debridement at the emergency room. Wound cultures were taken in 1187 patients (85.8%); the results showed that the positive rate of open fracture was 54.8% (651/1187), and 59% of the bacterial detections occurred in grade III fractures. Most pathogens (72.7%) were sensitive to prophylactic antibiotics, according to the EAST guideline. Quinolones and cotrimoxazole showed the lowest rates of resistance. The updated EAST guidelines for antibiotic prophylaxis in open fracture (2011) have been proven to be adequate for a large portion of patients, and we would like to suggest additional Gram-negative coverage for patients with grade II open fractures based on the results obtained in this setting in East China.

## 1. Introduction

The frequency of open fractures observed in any area varies according to geographical and socioeconomic factors, population size, and the system of trauma care. Open fractures are common in East China due to the rapid development of the manufacturing and transportation industries. As the infection rates of soft tissues and bones resulting from bacterial contamination have increased up to 50% [1], the incidence of complications, such as acute or delayed osteomyelitis, nonunion, or secondary amputation, may also increase. Although the factors contributing to infection after open fractures vary from person to person, infection prevention measures, including radical debridement and antibiotic prophylaxis, remain critical.

The current options for treating drug-resistant bacteria and bacterial infections include:(1)Antibiotics: Antibiotics are the most common treatment for bacterial infections. They work by killing or stopping the growth of bacteria. However, some bacteria have become resistant to antibiotics, making it difficult to treat the infections caused by these bacteria. In some cases, stronger antibiotics may be used to treat drug-resistant infections.(2)Combination therapy: Combination therapy involves using two or more antibiotics together to treat an infection. This can be helpful in treating drug-resistant bacteria, as it may be more effective than using a single antibiotic.(3)Probiotics: Probiotics are live bacteria and yeasts that are beneficial to the body. They can help restore the natural balance of bacteria in the gut, which can be disrupted by antibiotics. Probiotics can also boost the immune system and help fight off infections.(4)Antimicrobial stewardship: Antimicrobial stewardship is a coordinated effort to optimize the use of antibiotics and other antimicrobial drugs. This can involve educating healthcare providers on appropriate prescribing practices, monitoring the use of antibiotics, and implementing guidelines for the use of antibiotics in specific clinical situations.

The Eastern Association for the Surgery of Trauma (EAST) Practice Management Guidelines for prophylactic antibiotic use in open fractures were mainly based on literature emanating from the USA, Canada, Australia, Israel, South Africa, Ethiopia, and Saudi Arabia. No Asian studies were taken into consideration. However, pathogens and their resistance against antibiotic agents change with time and vary in different regions [2,3,4].

In this study, we aimed to characterize the bacterial spectrum present in open fractures, analyze the bacterial resistance to antibiotic agents, and examine the therapeutic regimes in trauma centers in East China.

## 2. Materials and Methods

The study protocol was approved by the Ethics Committee of the Shanghai Jiao Tong University Affiliated Sixth People’s Hospital. Informed consent was obtained from all participants. All study methods were in accordance with the Declaration of Helsinki.

The pathogens and antibiotic resistance patterns of open fractures in East China were analyzed using data collected as part of a multicenter study (an epidemiological investigation of traumatic infections of the upper and lower extremities) that was designed to analyze the pathogens, antibiotic resistance patterns, and risk factors for soft tissue and/or bone infection after an open fracture.

Six trauma centers in East China participated in this retrospective and prospective cohort study. Patients’ data from six of the seven states in East China were covered in this multicenter study.

The study protocol was approved by the institutional review boards of each of the centers. All of the patients consented to participate in the study, including follow-up evaluations.

We included all patients who sustained open fractures from January 2015 to December 2017 at six trauma centers. The patients’ data, as well as their Gustilo-Anderson classification, were noted, and the data were collected anonymously at the participating trauma centers with an Excel form and transferred to an SPSS chart. Bacterial contamination was assessed using deep tissue samples, and antibiotic resistance patterns were assessed using VITEK^®®^ 2 (BioMérieux, Marcy-l’Étoile, France).

## 3. Results

Each of the six centers included in this study was the largest trauma center in the state where it was located. In total, 1348 patients were included, with ages ranging from 16 to 82 years, with a mean of 37.5 years. Eight hundred and eighty-seven patients were male (64.1%).

The etiologies were mostly road traffic accidents (1107 patients, 80.0%). The lower leg (728 patients, 52.6%) and hand (519 patients, 37.5%) were the most commonly injured locations in this study. Most patients in this study had Gustilo-Anderson grade II open fractures (578 patients, 42.9%).

All patients received prophylactic antibiotics during the first debridement at the emergency room (Table 1). Microbiological samples were taken from the deep tissues of the fracture site in 545 patients (40.4%). Swabs taken from the wounds were cultivated in 339 patients (25.1%), and tissue samples were taken from 103 patients (7.6%). Most of the samples (558 samples, 56.5%) were drawn from patients with Gustilo-Anderson grade II open fractures. Intraoperative samples, or swabs, were collected from all patients who sustained Gustilo-Anderson grade III open fractures.

The results of all wound cultures were positive in 552 patients (55.9%). The positivity rates were 48.7% and 70.0% for grades II and III open fractures, respectively (Table 2). Coagulase-negative staphylococci were the most commonly isolated pathogens (37.5%) in grade II open fractures, and *Staphylococcus aureus* was the most commonly isolated pathogen (37.1%) in grade III open fractures. The other commonly isolated pathogens in grade II open fractures were *Staphylococcus aureus* (21.7%), *Pseudomonas aeruginosa* (9.9%), *Enterobacter cloacae* (7.7%), *Acinetobacter baumannii* (7.0%), *Escherichia coli* (5.9%), and group B streptococci (3.7%). The other commonly isolated pathogens in grade III open fractures included coagulase-negative staphylococci (23.7%), *Pseudomonas aeruginosa* (10.1%), *Acinetobacter baumannii* (8.6%), and *Escherichia coli* (6.8%). Six cases of grade III open fractures had polymicrobial wound contamination (2.2%). Anaerobic strains were found in one patient with grade II open fractures and three patients with grade III open fractures. (Table 3)

Most pathogens, except *Pseudomonas aeruginosa*, *Enterobacter cloacae*, and *Acinetobacter baumannii*, were sensitive to the prophylactic antibiotics according to the EAST guidelines [5]. We found Gram-negative strains in 83 (30.5%) of grade II open fractures and 84 (30.2%) of grade III open fractures. Among the Gram-negative strains, 21 strains were against gentamicin (12.6%) and 30 against ciprofloxacin (18.0%). We also found 95 problematic Gram-positive strains in grades II and III open fractures that were resistant to first-generation and second-generation cephalosporins and aminopenicillins. Twenty-one of these 45 strains (46.7%) were also not susceptible to gentamicin. (Table 4)

## 4. Discussion

Infections and complications remain major obstacles to the treatment of open fractures of the lower extremities [5,6,7,8]. Our multicenter study was a retrospective study based on six trauma centers in East China, including the Shanghai Jiao Tong University affiliated Sixth People’s Hospital (Shanghai, China), the Xiamen University affiliated Fuzhou Second Hospital (Fuzhou, China), the first affiliated hospital of Soochow University (Suzhou, China), the Tongde Hospital of Zhejiang Province (Hangzhou, China), the Second Hospital of Anhui Medical University (Hefei, China), and Shandong Provincial Hospital (Jinan, China).

To our knowledge, this is the first study focusing on the bacterial spectrum and resistance patterns in a cohort of patients with open fractures in East China. The recommended EAST antibiotic prophylaxis guidelines were published in 2000 on the EAST website. Based on a review of 54 articles published from 1975 to 1997, the workgroup offered three level I and two level II recommendations specific to the choice of antibiotic coverage and duration of therapy; the EAST guidelines were updated in 2011 [9,10]. Systemic antibiotic coverage directed at Gram-positive organisms and initiated as soon as possible after an injury is recommended. Additional Gram-negative coverage should be added for grade III open fractures. The EAST guidelines for antibiotic prophylaxis were followed at two of our trauma centers.

According to our results, the bacterial spectrum in the grade II open fractures differed from that of the grade III open fractures; in total, 37.5% of cultures revealed coagulase-negative staphylococci in grade II open fractures and 23.7% in grade III open fractures. The rate of Staphylococcus aureus infection increased from grade II to grade III open fractures. The overall incidences of infections with Gram-positive strains were higher than previously reported from European countries [11,12,13], which can be explained by the fact that four of our trauma centers did not follow the EAST guidelines. The major problem among these four trauma centers was the delayed administration of prophylactic antibiotics. Lack et al., found lower rates of infection when patients received antibiotics <66 min after injury [11]. No details regarding door-to-antibiotic administration time were available in our study, but they were mostly longer than 4 h according to trauma surgeons in these four centers.

Of significance, the incidence of Gram-negative infections has been increasing in China in the past few years. According to the results of our study, the incidences of Gram-negative bacilli infections in open fractures in grades II and III were around 30.5%. This incidence was relatively high when compared to that reported from Germany, which was as low as 11% [13]. These pathogens isolated from grade II open fractures would not have been adequately covered even if the EAST guidelines were followed, in addition to the other four trauma centers that did not follow the EAST guidelines. Among all the isolated Gram-negative bacteria, nosocomial strains, such as *Pseudomonas aeruginosa, Acinetobacter baumannii*, and *Enterococcus faecium*, had surprisingly high incidences (>20%). This may be attributable to several factors, including the limited trauma care workforce, the lack of well-established treatment protocols for severe open fractures, and the failure to strictly follow the guidelines regarding antibiotic prophylaxis. 

One patient with grade II open fractures and three patients with grade III open fractures were infected with anaerobic strains, possibly due to heavily soiled wounds. Heavily soiled wounds would be problematic in severe open fractures due to the potentially increasing load of anaerobic bacteria and the common use of cephalosporins with low levels of activity against anaerobic pathogens.

We found 45 problematic Gram-positive strains in grades II and III open fractures that were resistant to first- and second-generation cephalosporins and aminopenicillins. The lowest rates of resistance were identified among strains treated with quinolones, followed by cotrimoxazole. The fluoroquinolones were suspected of having a negative effect on bone healing; no advantage of fluoroquinolones was found compared with a combination regimen of a cephalosporin and an aminoglycoside recommended by the EAST guidelines [10,14,15]. Furthermore, the relatively small proportion prohibits general recommendations.

The positivity rates of the samples obtained from grade III open fractures were 70.0%, which was higher than that of the samples obtained from grade II open fractures. This result can be explained by the fact that high-velocity injuries often result in poor tissue oxygenation and the devitalization of soft tissue and bone. This produces a perfect medium for bacterial multiplication and infection. Accordingly, the greater the volume of involvement of the soft tissue bed in the injury, the easier it is for bacterial multiplication and infection to occur.

Our data highlighted the different characteristics of contamination based on the Gustilo-Anderson classification of open fractures. Wounds from grade III open fractures are more easily contaminated with Staphylococcus aureus compared to those from grade II open fractures. Antibiotic prophylaxis has to be effective against mostly Gram-positive bacteria. Nosocomial infections are mostly due to Gram-negative and drug-resistant strains and usually occur among patients on prolonged treatment with extended exposure times to wounds in trauma centers.

Due to the obstacles in the interhospital exchange of diagnosis and treatment information in China, it has become difficult to collect statistics on the strains of bacteria causing open fractures and their drug resistance. Therefore, it is necessary for the relevant national agencies to intervene and establish a database of information related to open fractures that can be updated in real time with the common strains and their drug resistance. Additionally, guidelines need to be developed and implemented in hospitals at all levels.

The major limitation of this study was the limited number of patients included and the even lower number of patients for whom microbiological samples were initially taken. Most of the trauma centers in our multicenter study lacked a standard protocol for antibiotic prophylaxis and wound sampling. In addition, some of the samples in our study (8%) were taken via swabs, which only represented the microbial colonization of the wound or surrounding skin rather than pathogens causing deep-tissue infections. A lower number of samples may cause difficulties in interpretation, and a higher number would lead to an increased probability of contamination without evidence of the improved sensitivity of the examination.

## 5. Conclusions

The spectrum of bacteria infecting patients with open fractures is changing in East China. The incidence of nosocomial infection seems to be increasing not only among patients with grade III open fractures but also among those with grade II open fractures. The updated EAST guidelines for antibiotic prophylaxis in open fracture (2011) have been proven to be adequate for a large portion of patients in the USA and most countries in Europe; they have also been proven to be clinically instructive in East China.

## Figures and Tables

**Table 1 jpm-13-00735-t001:** Prophylactic antibiotics according to the Gustilo-Anderson classification.

Gustilo-Anderson Classification	Prophylactic Antibiotics	*n*
**I**	Cefazolin	114 (30.6%)
	Cefuroxime	240 (64.3%)
	Clindamycin	19 (5.1%)
**II**	Cefazolin	187 (32.4%)
	Cefuroxime	306 (52.9%)
	Clindamycin	46 (8.0%)
	Cefuroxime/Metronidazole	24 (4.2%)
	Cefazolin/Metronidazole	15 (2.6%)
**III**	Cefazolin	21 (5.3%)
	Cefuroxime	195 (49.1%)
	Clindamycin	28 (7.1%)
	Gentamicin	31 (7.8%)
	Cefuroxime/Metronidazole	82 (20.7)
	Cefazolin/Metronidazole	40 (10.1%)

**Table 2 jpm-13-00735-t002:** The number of open fractures, microbiological samples, and positive results according to the Gustilo-Anderson classification.

Gustilo-Anderson Classification	Number of Patients	Number of Microbiological Samples	Number of Isolated Pathogens
**I**	373	32	2
**II**	578	558	272
**III**	397	397	290 *

* Polymicrobial wound contamination was found in six patients with III open fractures, which included: *Staphylococcus epidermidis*/*Pseudomonas aeruginosa*, *Staphylococcus capotis*/*Acinetobacter baumannii*, *Staphylococcus epidermidis*/*Escherichia coli*, *Pseudomonas aeruginosa*/*Enterococcus faecium*, *Staphylococcus aureus*/*Acinetobacter baumannii*, and *Staphylococcus aureus*/*Enterobacter cloaca*.

**Table 3 jpm-13-00735-t003:** The number of isolated pathogens according to the Gustilo-Anderson classification.

	COST *	*Staphylococcus Aureus*	*Pseudomonas Aeruginosa*	*Enterobacter Cloacae*	*Acinetobacter Baumannii*	*Escherichia coli*	*Streptococcus B*	*Enterococcus Faecium*	*Corynebacterium Jekeium*	Anaerobic Strains **	Others ***
**I**	2	0	0	0	0	0	0	0	0	0	0
**II**	102	59	27	21	19	16	10	9	7	1	1
**III**	66	103	28	13	24	19	13	13	6	3	2

* COST: coagulase-negative staphylococci. ** Anaerobic strains: clostridium perfringens. *** Others include *Klebsiella pneumonia* and *Pseudomonas stutzeri.*

**Table 4 jpm-13-00735-t004:** Resistance of the isolated pathogens according to the Gustilo-Anderson classification.

Gustilo–Anderson Classification	Isolated Pathogens	*n*	Resistance	*n*
**I**	*Staphylococcus epidermidis*	2	-	-
**II**	*Staphylococcus epidermidis*	67	Cefazolin	23
			Cefuroxime	28
			Gentamicin	5
			Penicillin	34
			Cotrimoxazole	29
			Amoxicillin Clavulan acid	21
			Tetracycline	27
	*Staphylococcus capitis*	35	Penicillin	16
			Gentamicin	35
			Cotrimoxazole	35
			Oxacillin	35
			Clindamycin	35
			Fosfomycin	9
	*Staphylococcus aureus*	59	Penicillin	11
			Cefazolin	16
			Cefuroxime	18
			Levofloxacin	8
			Ciprofloxacin	9
			Gentamicin	3
			Clindamycin	4
			Moxifloxacin	6
			Tetracycline	2
			Amoxicillin clavulan acid	4
	*Pseudomonas aeruginosa*	27	Cefperazone–sulbactam	11
			Gentamicin	2
			Levofloxacin	6
			Ciprofloxacin	6
			Imipenem	13
			Cefepime	19
			Cefazolin	19
			Cefuroxime	14
	*Enterobacter cloacae*	21	Imipenem	16
			Aztreonam	18
			Cefperazone–sulbactam	6
			Ceftazidime	19
			Cefepime	18
			Gentamicin	7
			Ceftriaxone	20
			Levofloxacin	9
	*Acinetobacter baumannii*	19	Ciprofloxacin	14
			Moxifloxacin	15
			Ceftazidime	15
			Imipenem	13
			Cefepime	16
			Ceftriaxone	16
			Cotrimoxazole	11
	*Escherichia coli*	16	Cefuroxime	12
			Amoxicillin clavulan acid	11
			Imipenem	9
			Ceftazidime	13
			Cefepime	13
			Cefperazone–sulbactam	15
			Ceftriaxone	16
			Gentamicin	5
			Ampicillin	10
	Streptococcus B	10	Tetracycline	5
			Gentamicin	3
	*Enterococcus faecium*	9	Cefuroxime	9
			Gentamicin	4
			Ciprofloxacin	9
			Moxifloxacin	9
			Cotrimoxazole	1
			Rifampicin	9
			Vancomycin	1
			Tetracycline	3
			Levofloxacin	2
			Ampicillin	1
	*Corynebacterium jekeium*	7	Ceftazidime	7
			Penicillin	7
			Cefepime	7
			Fosfomycin	6
			Ciprofloxacin	6
			Moxifloxacin	6
			Cotrimoxazole	1
	*Clostridium perfringens*	1	Penicillin	1
			Aztreonam	1
			Amoxicillin clavulan acid	1
			Tetracycline	1
			Piperacillin	1
			Azithromycin	1
	*Klebsiella pneumonia*	1	Imipenem	1
			Tigecycline	1
			Cefperazone–sulbactam	1
			Levofloxacin	1
**III**	*Staphylococcus epidermidis*	45	Cefazolin	26
			Cefuroxime	26
			Gentamicin	3
			Penicillin	38
			Amoxicillin clavulan acid	16
			Tetracycline	39
			Cotrimoxazole	35
	*Staphylococcus capitis*	21	Oxacillin	21
			Penicillin	15
			Fosfomycin	7
			Gentamicin	21
			Cotrimoxazole	21
			Clindamycin	21
	*Staphylococcus aureus*	103	Penicillin	39
			Cefazolin	22
			Cefuroxime	22
			Levofloxacin	16
			Ciprofloxacin	17
			Gentamicin	7
			Clindamycin	6
			Moxifloxacin	16
			Vancomycin	2
			Tetracycline	14
			Amoxicillin clavulan acid	11
	*Pseudomonas aeruginosa*	28	Cefperazone–sulbactam	13
			Piperacillin–yazobactam	1
			Gentamicin	3
			Levofloxacin	7
			Ciprofloxacin	7
			Imipenem	14
			Cefepime	20
			Cefazolin	20
			Cefuroxime	19
	*Enterobacter cloacae*	13	Imipenem	3
			Aztreonam	5
			Cefperazone–sulbactam	6
			Ceftazidime	11
			Cefepime	11
			Ceftriaxone	12
			Levofloxacin	5
	*Acinetobacter baumannii*	24	Ciprofloxacin	3
			Ceftazidime	12
			Cefepime	12
			Ceftriaxone	12
			Cotrimoxazole	4
	*Escherichia coli*	19	Cefuroxime	13
			Amoxicillin clavulan acid	13
			Levofloxacin	2
			Ceftazidime	13
			Cefepime	13
			Cefperazone–sulbactam	19
			Ceftriaxone	15
			Gentamicin	4
			Ampicillin	11
	Streptococcus B	13	Tetracycline	8
			Gentamicin	4
	*Enterococcus faecium*	13	Cefuroxime	13
			Ciprofloxacin	13
			Moxifloxacin	13
			Cotrimoxazole	1
			Ampicillin	1
			Rifampicin	13
			Vancomycin	1
			Tetracycline	4
			Levofloxacin	2
	*Corynebacterium jekeium*	6	Ceftazidime	6
			Cefepime	6
			Fosfomycin	6
			Ciprofloxacin	6
			Moxifloxacin	6
			Cotrimoxazole	1
	*Clostridium perfringens*	3	Penicillin	3
			Aztreonam	3
			Amoxicillin clavulan acid	3
			Tetracycline	3
			Piperacillin	3
			Azithromycin	3
	*Klebsiella pneumonia*	1	Imipenem	1
			Tigecycline	1
			Levofloxacin	1
			Cefperazone–sulbactam	1
	*Pseudomonas stutzeri*	1	Ceftazidime	1
			Cefperazone–sulbactam	1
			Ceftriaxone	1
			Levofloxacin	1

## Data Availability

Data are available upon request due to restrictions on privacy. The data presented in this study are available upon request from the corresponding author.

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
