# Peer review of "Bacterial Contamination of Open Fractures: Pathogens and Antibiotic Resistance Patterns in East China"

_jpm, 2023, doi:10.3390/jpm13050735_

Round 1

Reviewer 1 Report

1. Introduction, clinically, some of the conventional ways of treating drug-resistant bacteria and bacterial infections can be described

2. Do some emerging antibiotics, such as metformin, also lead to bacterial resistance?

3. Are hydrogel dressings (10.1021/acsmacrolett.2c00290; 10.1016/j.indcrop.2022.115273) also an option for clinical treatment of bacterial resistance?

4. It could be better if a brief comment (challenges and future prospects) is added at the manuscript.

5. The Abstract section should be clear and concise: the important results and main conclusions drawn in this paper should be highlighted and presented in more precise language.

Author Response

  1. Introduction, clinically, some of the conventional ways of treating drug-resistant bacteria and bacterial infections can be described

Thanks for your suggestion, the description has been added to introduction part. 

  1. Do some emerging antibiotics, such as metformin, also lead to bacterial resistance?

Thanks for your comments, since the drug-resistance test haven’t include those emerging antibiotics, we can’t tell if they already lead to bacterial resistance.

  1. Are hydrogel dressings (10.1021/acsmacrolett.2c00290; 10.1016/j.indcrop.2022.115273) also an option for clinical treatment of bacterial resistance?

 Thanks for your comments, there were fundamental research on the bacterial resistance effect of hydrogel dressing. But no reliable clinical RCT has been published yet.

  1. It could be better if a brief comment (challenges and future prospects) is added at the manuscript.

 Thanks for your suggestion, a brief comment about challenges and future prospects has been added.

  1. The Abstract section should be clear and concise: the important results and main conclusions drawn in this paper should be highlighted and presented in more precise language.

Thanks for your advice, the Abstract section has been revised.

Reviewer 2 Report

This retrospective investigation in China addresses the complex scenario of open fractures. In addition, the study included a significant number of individuals, only seen in trauma centers. However, there is a severe problem distinguishing between contamination and infection, in addition to divergences between the abstract and the text and a need for more basic information (time of surgical approach, number of samples collected, whether a deep swab of exudate was performed).

Antibiotic changes cannot be based on swabs or tissue culture on admission, as the pathogens involved in a later infection differ. In addition, there was no clear definition of the time of collection and whether there was an infection (another point: there is no information on clinical signs and symptoms of FRI; the question about the time of collection remains).

Author Response

Thanks for your comments, hand injury was also classified to GA classification

We presumed different geographical regions and hospitals contribute to microbiological profiles, and further investigation was conducted to prove our speculation.

Yes, cultures were collected on hospital admission.

Data were collected by 6 statisticians who unaware of the study.

987 samples totally. All from soft tissue. Swabs taken from the wounds were cultivated in 339 patients.

Resistant against gentamicin and ciprofloxacin were found for Staphylococcus aureus and some other pathogens.

Reviewer 3 Report

Reviewers’ comments for the Manuscript ID: jpm-2185793

The manuscript title:Bacterial contamination of open fractures: Pathogens and anti- 2 biotic resistance patterns in East China”

In the current manuscript, authors report the Identification of different bacterial spectrum in open fractures reported in in six major trauma centers in East China from January 2015 to December 2017. Although, authors described differential expressions of microorganisms in grade I to III and their resistance to different antibiotics, but supporting data for authors claims are completely missing, it is poorly, written manuscript, many grammatically errors and very difficult to read, so manuscript, should be completely revised before publication.

Author Response

Thanks for your suggestions, the manuscript has been revised by a native speaker.

Round 2

Reviewer 2 Report

Antimicrobial prophylaxis should follow isolated pathogens in cases of infection. However, what the authors propose (conclude) goes against the evidence: typically, the microorganism isolated on admission is not the causative agent of the infection related to the fracture. Therefore, the authors should strongly consider changing the conclusion (and discussion excerpts). 

Excerpt from conclusion under discussion: "... we would like to suggest additional Gram-negative coverage for patients with grade II open fractures based on the results...".

Author Response

Excerpt from conclusion under discussion: "... we would like to suggest additional Gram-negative coverage for patients with grade II open fractures based on the results...".

Thank for your advice, the nosocomial strains s not the causative agent of the infection related to the fracture. the conclusion part has been revised. 

Reviewer 3 Report

Reviewers’ comments for the Manuscript ID: jpm-2185793

The manuscript title:Bacterial contamination of open fractures: Pathogens and antibiotic resistance patterns in East China”

Revised manuscripts, reads well and may be suitable for publication, still below few comments needs to be addressed before publication.

Comments

all the antibiotic drugs names in table 1, table 4 and throughout the manuscript should start with capital letters.

Author Response

Comments

all the antibiotic drugs names in table 1, table 4 and throughout the manuscript should start with capital letters.

Thanks for your suggestions, the names has been revised in tables